# Effects of New Seal Introductions on Conspecific and Visitor Activity

**DOI:** 10.3390/ani12212962

**Published:** 2022-10-28

**Authors:** Emily J. Northey, Baine B. Craft, Eduardo J. Fernandez

**Affiliations:** 1Department of Research Psychology, Seattle Pacific University, Seattle, WA 98119, USA; 2School of Animal & Veterinary Sciences, The University of Adelaide, Adelaide, SA 5005, Australia

**Keywords:** animal introductions, animal-visitor interactions, seal activity, visitor activity, zoo welfare, harbor seals, northern fur seals

## Abstract

**Simple Summary:**

Modern zoos and aquariums are defined by several goals, which include improving the welfare of their animals, conservation efforts, public education, research, and visitor enjoyment. Animal-Visitor Interactions (AVIs) are a means to study the outcomes of some of these goals. These outcomes can be measured in terms of visitor effects (the effects of the visitors on the exhibited animals) and visitor experiences (the effects of the animals and the institution itself on the visitors). Additionally, new animal introductions have been examined for their effects on animals, but little has been done to measure the introduction visitor experience. The purpose of this study was to observe the effects of new animal introductions at two exhibits on both animal and visitor activity. Changes in existing animals’ behaviors and visitor crowd size and length of stay were measured before and after new animal introductions. The introduction had neutral to positive impacts on both variables, suggesting potential benefits on AVIs based on new animal introductions. Such studies could help establish best practices for new animal introductions and ways to promote positive AVIs within a zoo/aquarium setting.

**Abstract:**

Modern zoos and aquariums are defined by several primary goals, which include improving the welfare of their animals, public education, and inspiring visitors to develop an interest in conservation. Animal-Visitor Interactions (AVIs) (i.e., the impact of captive animals and visitors on each other) are a primary means to study these goals. The introduction of new animals into two seal exhibits—the harbor and northern fur seals—at the Seattle Aquarium presented a valuable opportunity to assess the impacts of introductions on the currently exhibited animals and aquarium visitors. The impacts of animal introductions were measured through direct observations of seal activity (i.e., ethogram behavioral observations) and visitor activity (i.e., crowd size and visitor length of stay) before and after new seal introductions. This study consisted of two experiments: Experiment 1 (Harbor seal exhibit) and Experiment 2 (Northern fur seal exhibit). In Experiment 1, we found that the introduction had few impacts on the existing harbor seals or on visitor activity. In Experiment 2, the introduction of a new fur seal had significant positive impacts on the existing fur seal, as was seen through increased social behaviors and decreased stereotypic behaviors, as well as a significant increase in crowd size post-introduction. Based on these findings, new animal introductions were associated with neutral to positive changes in both seal and visitor activity. Findings from this study suggest that studying new animal introductions can result in better understanding and promoting positive interactions with existing animals (introduction effects) and likewise positive experiences for zoo and aquarium visitors (introduction experiences).

## 1. Effects of New Seal Introductions on Conspecific and Visitor Activity

Animal-Visitor Interactions (AVIs) are a popular area of research interest within zoo and aquarium studies [1]. AVIs—which involve the ways in which captive animals and visitors can impact each other’s behaviors—are in line with the goals of modern zoos and aquariums. Zoos and aquariums are interested in cultivating positive experiences among the visitors as a way to invoke empathy and positive attitudes towards the animals, as well as generating further support for their facilities and their conservation efforts [2,3,4,5]. It is sometimes the case that visitors are drawn to zoos and aquariums due to an interest in observing animals that are exotic and rare to the geographical area. Additionally, educational efforts in zoos and aquariums can be an effective method to generate interest in environmental issues, such as conservation and biodiversity [6,7].

Zoos and aquariums attempt to meet many of their goals, including education and conservation, by introducing visitors to informative and engaging exhibits and experiences. These interactions can range from physical interactions, to viewing the animals, to listening to an educational talk. For instance, D’Cruze et al. [8] reviewed the variety of AVI activities that occur at zoos and aquariums worldwide, finding that handfeeding, riding wild animals, and walking or swimming alongside animals were the most popular. Previous research has shown that when visitors are able to interact with the animals at a zoo or aquarium, visitors express an increased concern, knowledge, and appreciation for those species [7,9,10,11,12]. This was often seen through expressing interest in conservation efforts, increased attendance, and financial donations from visitors [7,13,14].

Nonetheless, the goals of modern zoos and aquariums can be in conflict, specifically between animal welfare and visitor interactions. For example, allowing visitors to get close to animals can increase visitor interest and may even have a positive impact on some animals who are interested in interacting with visitors, but on the other hand, large crowd sizes and a high frequency of visitors may be detrimental to the welfare of the animals [2,5,15,16,17]. However, without the ability for visitors to closely observe and interact with the animals, there can be missed opportunities for developed interest among visitors. Additionally, a lack of visitor interest can lead to decreased financial support of zoos and aquariums [18,19]. Therefore, the overarching focus of zoos and aquariums is to promote a positive experience for both the animals and the visitors.

AVIs can be studied in two ways: visitor effects and visitor experiences. *Visitor effects* are defined as the ways in which visitors impact the behaviors of animals [16]. This has been commonly measured through animals’ hormone levels, spatial dispersion, visibility within an enclosure, and behavioral observations of the animals [16,20,21,22]. An increased visitor presence could elicit a positive, neutral, or negative impact on the currently exhibited animals [5,16,18]. For instance, positive visitor effects could be seen through greater general activity, such as increased foraging and social interactions with other animals [16,23]. It is also important to consider how negative visitor effects impact the welfare of the animals. This has been seen in previous studies where an increase in visitor frequency and crowd size led to a decrease in activity by the zoo animals [24,25]. Additionally, animals may become habituated to the presence of visitors and there is little to no effect of visitors on animal behavior, which would be categorized as a neutral visitor effect [16,26]. Furthermore, enclosure usage can be incorporated as a measure of behavioral welfare for captive animals, particularly when it is difficult to derive meaningful information from behaviors alone [20,22]. *Visitor experiences*, on the other hand, are defined as the ways in which animals, exhibits, and zoos/aquariums in general impact visitors. Visitor experiences can be measured directly through behavioral measures such as crowd size or visitor length of stay [26,27], as well as through indirect measures such as surveys of visitor perceptions [4,28].

### 1.1. Animal Introductions and Animal-Visitor Interactions

How animals are housed socially—including the introductions of new animals—are important factors for the welfare of those exhibited animals [29,30,31,32]. Multiple studies have directly examined the effects of introducing new animals to a zoo exhibit [33,34,35,36,37,38,39]. In addition, other researchers have experimentally examined introduction effects of zoo animals on currently exhibited conspecifics, where the ways in which animals are housed together are repeatedly and systematically manipulated. For example, Fernandez and Harvey [22] examined enclosure use variability (entropy) in African wild dogs (*Lycaon pictus*) when one of the dogs was removed periodically. Rowden [40] manipulated the grouping of Bulwer’s wattled pheasants (*Lophura bulweri*), where pheasants were systematically removed from six to three individuals. Similarly, other researchers have examined the impact of changes in how animals are housed together within African elephants (*Loxodonta africana*) [41], giraffes (*Giraffa camelopardalis*) [42,43,44], and black and gold howler monkeys (*Alouatta caraya*) [45]. Finally, Luebke et al. [13] examined how zoo visitors answered questions related to viewing animal social interactions.

New animal introduction effects are multifaceted, impacting both animals and visitors alike. Thus, the study of new animal introductions on animals and visitors can be thought of like that of visitor effects and experiences, where *introduction effects* examine the potential impacts of a new animal introduction on the animals that are currently housed in the enclosure, and *introduction experiences* examine the potential impacts of a new animal introduction on the visitors. While the studies above have focused on social housing and new animal *introduction effects*, we know of no published studies to date that have examined *introduction experiences*. Thus, the study of how new animal introductions impact the visitors themselves should play a more important role in our understanding of both introductions and AVIs.

### 1.2. Pinniped Natural History and Sociality

The aggregation of seals in the wild is likely due to limitations in resources, such as areas to haul-out and reproduce that are safe from predation, and food resources [46,47,48,49]. Harbor seals (*Phoca vitulina*) and northern fur seals (*Callorhinus ursinus*) primarily interact with other conspecifics when hauled-out on land [50,51,52]. Harbor seals are often seen resting on land in herds and display aggressive behaviors towards one another in order to fight for landing and resting areas, rather than for access to females [47,51,53,54]. Access to space is typically awarded through dominance ranking that is based on size and sex (i.e., larger adult males) [54]. In contrast, when northern fur seals are hauled out on land together, this tends to result in male-male competition for access to females during breeding season [47,51,53]. During this time, males become territorial and will fight with other males in order to establish dominance within their hierarchical social structure and gain access to females, which is typically awarded based on size and sex (i.e., larger adult males) [47,51,53].

These social behaviors and social structures have been seen among captive and wild pinnipeds alike [55]. In captivity, harbor seals tend to engage with their environment more so than with other individuals in their exhibit, while northern fur seals typically engage more with other conspecifics, and less so with their environment [55,56]. Age and relatedness also impact the frequency of interactions between conspecifics, where older seals are less likely to interact with others [55,56]. A recent study by Jenkins [55] found that individuals spend more time with other seals that they are related to (i.e., a mother-son pair) rather than unrelated seals. Additionally, personality differences can influence the social structure within captive settings and should be taken into consideration when housing individuals together [56].

Another impact of social housing that is important to consider is single-sex versus mixed-sex group housing. Previous studies have shown that seals that are housed in enclosures with other members of the same sex tend to exhibit more social behaviors and interact with each other more often than when they are housed with members of different sexes [55,57]. A recent study by Meyer et al. [57] found that when male California sea lions were housed with other males at a rehabilitation facility, the individuals displayed more social behaviors, such as interacting with others and swimming together. This level of activity was consistent throughout the day within single-sex enclosures, whereas individuals that were housed in a mixed-sex enclosure had varied levels of activity, which decreased throughout the day [57]. By understanding how harbor seals and northern fur seals behave and interact with each other, findings from the present study can indicate best practices for social housing in zoos and aquariums that promote positive animal welfare.

### 1.3. Study Purpose and Predictions

The purpose of our study was to examine the impact of new animal introductions with two pinniped species (northern fur and harbor seals). We examined whether the introductions of a new individual impacted (a) the behaviors of the existing exhibited animal(s) (introduction effects) and (b) the behaviors of the aquarium visitors viewing the seal exhibits (introduction experiences). Introduction effects were observed through a set of 15 behaviors (behavioral ethogram) measured for the exhibited animals before and after a new seal introduction (pre- and post-introduction), while introduction experiences were measured through pre- and post-introduction changes in crowd size and visitor length of stay. We predicted that the introductions could have positive, neutral, or negative impacts for the exhibited seals and visitors, as observed through changes in seal or visitor activity, respectively. We expected that the introductions were more likely to have neutral to positive effects on both seal and visitor activity, with positive effects suggesting that the new seal introductions benefited both the welfare of the exhibited animals as well as improved the visitor experience.

## 2. General Methods

### 2.1. Subjects and Settings

This research study was conducted at the Seattle Aquarium in Seattle, WA. The Seattle Aquarium, which opened in 1977, is located on Piers 59 and 60 on Elliott Bay. Two different pinniped species were observed in this study: three male harbor seals (*Phoca vitulina*) (namely, Barney, Hogan, and Casey) and two male northern fur seals (*Callorhinus ursinus*) (namely, Flaherty and Chiidax). Specifics about each seal and their exhibits are detailed within each experiment’s description below.

Enrichment devices were frequently placed in the exhibits such as plastic balls, buoys, hard hats, and buckets. The seals were fed three to four times per day between approximately 07:00 and 16:00. Their diet consisted of sustainably caught seafood such as herring, capelin, mackerel, squid, anchovies, eulachon smelt, and salmon. Small baitfish such as whitebait, bird herring, silversides, and sand eels were given as a part of enrichment activities (these food items were not alive when fed to the seals). Marine mammal presentations were given periodically by aquarium staff at each exhibit, which involved giving information about the individual seals and their species (e.g., their diet, wild habitat, and hunting abilities), promoting conservation, as well as answering audience questions.

### 2.2. Materials

Paper data sheets were used to collect animal and visitor data by hand for each session. Clipboards and pens were used as writing material. Researchers used the mobile app MultiTimer: Multiple Timers (MultiTimer) [58] to signal various time-points to record data (i.e., a one min timer to record seal behaviors, a 10 min timer to record crowd size, and a stopwatch to count visitor length of stay). The mammals’ behaviors were categorized based on a behavioral ethogram that was developed and modified from de Vere et al. [56], Fernandez & Timberlake [59], and Island et al. [60] (see Table 1). The behavioral ethogram was comprised of 15 behaviors split into five major classes: Active, Inactive, Social, Stereotypy and Other. Informational signs were placed within the exhibit boundaries to notify visitors of the ongoing research study and provided contact information of the primary investigator should they have additional questions or concerns. Other information on the new seal introductions included press releases from the Seattle Aquarium about the new animals, interviews by Seattle Aquarium personnel with local news stations, Seattle Aquarium blog posts, and updates on the aquarium website and their social media accounts. Additionally, aquarium guests were informed about the new animal introductions through volunteers and staff interpreting at the seal exhibits. Staff and volunteers (of which there are many throughout the building) were given information about the seals so they could be prepared to answer questions and encourage guests to visit the new animals.

### 2.3. Data Collection and Procedure

Prior to its implementation, the study was approved by the Seattle Aquarium Conservation Research Committee. The project received exemption from Seattle Pacific University’s (SPU) Institutional Animal Care and Use Committee (IACUC), and was approved by SPU’s Institutional Review Board (IRB #212209001, expiration 1/26/2023). Data collection ran from January 2022 to April 2022, during which the introductions for each of the seals occurred. Data collection was divided into Experiment 1 (harbor seals) and Experiment 2 (northern fur seals). Researchers collected data 7 days a week, between 09:00 and 17:00 (the aquarium was open to the public from 09:30 to 18:00 every day). Each session was considered a 1 h period of observation, with one to two sessions collected by each observer daily. Due to the timing of the construction for the Northern fur seal exhibit, there was brief overlap between the data collection period for both experiments (see exact dates and session information listed under each experiment’s methods).

The animal and visitor variables were measured simultaneously during each observation session. During each experiment, the behaviors of the animals in each setting were recorded prior to and after the introduction of the new individuals (pre-introduction and post-introduction conditions, respectively). These data were recorded using written datasheets where instantaneous sampling [20,61] was used to record the behaviors. Researchers used the phone app MultiTimer that sounded a beep every minute, signaling the researchers to write down the behavior that the animal(s) were doing in that exact instant. Each observation session lasted for 1 h with a total of 60 points of animal observation within each session. Results were later compared to determine if the introduction had a significant effect on the frequency of behaviors for the previously exhibited animals.

This methodology similarly applies to the visitor activity variables and was adapted from Godinez et al. [27]. Both crowd size and visitor length of stay were measured every 10 min, therefore providing six sample points for both of these measures for each 1 h observational session. Crowd size was measured by counting the number of individuals present within the exhibit boundary area every 10 min, which was signaled by a timer in the MultiTimer app. For the purposes of our study, an individual was considered any human in the exhibit, regardless of age. Length of stay was sampled by selecting a random visitor to measure the time they entered and exited the exhibit space using a stopwatch on the MultiTimer app. Specifically, every 10 min, researchers started coding the first adult (as estimated by the researchers) that walked into the exhibit area, recording the length of time they remain inside the exhibit boundaries, while also counting the crowd size. An individual’s length of stay was recorded for up to 10 min, and if they remained in the exhibit area for longer, their time would be listed as “10+”.

#### Overall Visitor Attendance

A total visitor attendance count at the Seattle Aquarium was gathered during the time of the study (see Figure 1A,B). An independent samples t-test was conducted to compare visitor attendance pre- and post-introductions for both Experiment 1 and 2. For Experiment 1, the pre-introduction visitor counts were gathered from 24 January 2022, to 7 February 2022. No visitor attendance data were collected on February 3rd, when the aquarium was closed to visitors for a special event. The post-introduction visitor counts were gathered from 8 February 2022, to 1 March 2022. For Experiment 2, the pre-introduction visitor counts were gathered from 25 February 2022, to 19 March 2022. The post-introduction visitor counts were gathered from 30 March 2022, to 30 April 2022. No visitor counts were gathered during the time that Chiidax was in the exhibit by himself (20 March 2022, to 29 March 2022).

### 2.4. Statistical Analyses

All comparisons were based on a within-subjects study design, with observations occurring per each seal and on a session-to-session basis [62]. IBM SPSS Statistics, Version 28.0 was used to run the statistical analyses for this project, with each session functioning as a sample point within each condition. For both Experiments 1 and 2, comparisons were conducted for the pre- and post-introduction conditions. All comparisons failed Shapiro–Wilk tests for normality or Levene’s tests for equality of variances, except three: overall visitor attendance during both periods (Experiment 1 and 2), and Experiment 1′s visitor length of stay. Therefore, a parametric independent sample t-test was used in these three instances. For all other comparisons, Mann–Whitney U tests were used. In addition, each experiment had a Bonferroni correction (Experiment 1: 12 comparisons, α= 0.004; Experiment 2: 7 comparisons, α = 0.007).

## 3. Experiment 1—Harbor Seals

Experiment 1 sought to evaluate the effects that a new seal introduction would have on the two previously exhibited harbor seals (*Phoca vitulina*), as well as on visitor activity, as described in Section 2.

### 3.1. Materials and Methods

Three captive-born harbor seals were the subjects of Experiment 1: Barney, Hogan, and Casey (see Figure 2). Barney, ~92.4 kgs, was a 36-year-old male who was born at the Seattle Aquarium. Hogan, ~78.7 kgs, was a 9-year-old male who originally came from the Point Defiance Zoo in Tacoma, Washington. Casey, ~94.0 kgs, was an 8-year-old male who came from Seaside Aquarium in Seaside, Oregon, where he was housed with 10 other harbor seals; three male and seven female. During the breeding season, the males would be separated from the females in order to prevent breeding. The pre-introduction condition consisted of observing Barney and Hogan, who were previously in the exhibit. During the post-introduction condition, an additional harbor seal, Casey, was added to the exhibit. Casey did not to undergo a quarantine period based on risk assessments done by the Seattle Aquarium veterinarian and curatorial staff and was allowed to be introduced directly into the harbor seal enclosure. All three seals were observed during the post-introduction condition.

The Harbor seal exhibit was located outside and overlooked Elliott Bay (see Figure 3). There were multiple signs around the exhibit where visitors could learn more information about the seals (e.g., lifespan, diet, and natural habitat). The enclosure was comprised of a saltwater pool and several dry haul-out spaces for animals to rest. The dry resting area/haul-out space was 19.5 m^2^. The saltwater pool was a depth of 1.83 m, and the volume was 147,929.77 L. The water was from Puget Sound, which undergoes sand filtration and was initially pumped into the largest fish exhibit at the Seattle Aquarium called the Underwater Dome. Some of that water was then UV sterilized and gravity fed into the Harbor seal habitat. Visitors can view the animals through two acrylic viewing panels outside—which allows for above water and underwater viewing—and one acrylic viewing panel inside—that allows for above water viewing and a direct view of one of the dry haul-out spaces.

#### Data Collection and Procedure

Data collection for Experiment 1 ran from 24 January 2022 to 1 March 2022 with a total of 94 observation sessions. Experiment 1 was divided into two conditions: pre-introduction and post-introduction. The pre-introduction condition ran from 24 January to 7 February and consisted of 43 observation sessions. The pre-introduction condition acted as a baseline condition for the two existing harbor seals in the exhibit (Barney and Hogan) to record their behaviors according to the behavioral ethogram. The post-introduction condition ran from February 8th to March 1st and consisted of 51 observation sessions. The post-introduction condition began when the new harbor seal (Casey) was introduced into the exhibit. Researchers recorded the behaviors of all three seals during this time according to the behavioral ethogram detailed in Section 2.

### 3.2. Results and Discussion

An activity budget for all three harbor seals displays the percentages that each seal engaged in each of the five classes of behavior following the new animal introduction (see Figure 4). There were no statistically significant differences in the five classes of behavior for the previously exhibited harbor seals as a result of the new seal introduction, aside from two: Barney’s Active behavior significantly decreased (*U*_43,51_ = 597.50, *p* < 0.001, *d* = 0.85), and his Inactive behavior significantly increased (*U*_43,51_ = 712.50, *p* = 0.004, *d* = 0.63), following Casey’s introduction (see Figure 5). There were no statistically significant differences in Hogan’s behaviors following the new seal introduction (see Figure 6).

There were no statistically significant differences in visitor crowd size or visitor length of stay as a result of the new seal introduction (see Figure 7A,B). Additionally, during Experiment 1, there was a significant increase in the overall visitor attendance in the post-introduction condition compared to the pre-introduction condition (*t*_34_ = −2.26, *p* < 0.05, *d* = −0.77; see Figure 1A).

Few changes were observed as a result of the introduction. The changes seen in Barney’s Active and Inactive behaviors were statistically significant but minor, suggesting that the introduction was not ideal for Barney. However, it did not appear to seriously impact his welfare in a negative way. The differences in Stereotypic behaviors for both Barney and Hogan were not statistically significant, further suggesting that the introduction did not negatively impact the previously exhibited seals’ welfare. Furthermore, the non-significant increase in crowd size that was observed from the pre-introduction condition to the post-introduction condition was likely due to the increase in overall visitor attendance at the aquarium, not a result of increased visitor interest in the harbor seals due to the new seal introduction. The significant increase observed in overall visitor attendance from the pre- and post-introduction conditions was likely due to seasonality, as Experiment 1 occurred during the transition from winter to spring. In conclusion, Casey’s introduction had few effects on the existing harbor seals—which is suggestive of neutral animal welfare—and any change in crowd size that was observed at the Harbor seal exhibit was predicted by an increase in overall visitor attendance, as seen in the significant change in visitors to the aquarium (see Figure 1A).

## 4. Experiment 2—Northern Fur Seals

Experiment 2 sought to evaluate the effects that a new seal introduction had on the one previously exhibited northern fur seal (*Callorhinus ursinus*), as well as on visitor activity, as described in Section 2.

### 4.1. Materials and Methods

Two northern fur seals were the subjects of Experiment 2: Flaherty and Chiidax (see Figure 8). Flaherty, ~123.4 kgs, was a 10-year-old male who was born at the New England Aquarium in Boston, Massachusetts. He was transferred to the Seattle Aquarium when he was 3 years old to be a companion for Leu, who passed away in 2020. Chiidax, ~84.8 kgs, was a 9-year-old male who came from the New England Aquarium, where he was housed with five other seals including three California sea lions; two adult females and one young male, and two northern fur seals; two young females. Chiidax was found as a pup in a box on the doorsteps of the Department of Fish and Wildlife in Alaska. Due to his young age, he was deemed unable to return to the wild and was raised at the New England Aquarium. In Experiment 2, the pre-introduction condition consisted of observing Flaherty alone. In between the pre- and post-introduction conditions, Chiidax went through a quarantine period where he was alone in the primary enclosure in order to ensure that he was healthy and acclimating well in his new environment following his transfer. As male northern fur seals of this age are naturally very territorial, it was important that Chiidax learned his new environment, demonstrated that he understood how to navigate the rockwork, and was eating well, prior to introducing Flaherty back into the exhibit with him. Data for this time period in between conditions was not analyzed. In the post-introduction condition, Flaherty was moved back into the primary exhibit with Chiidax.

The Northern fur seal exhibit was located inside; however, there was an open roof above the exhibit (see Figure 9). There were multiple signs around the exhibit where visitors could learn more information about the seals (e.g., lifespan, diet, and natural habitat). The enclosure was comprised of a saltwater pool, several dry haul-out spaces for animals to rest, and a rock wall for the fur seals to climb. The dry resting area/haul-out space was 40.51 m^2^. The saltwater pool was a depth of 3.66 m, and the volume was 295,495.85 L. The water for the saltwater pool came from the Puget Sound and underwent sand filtration and was pumped into the fur seal habitat. There was also additional water filtration through sand filters. The animals could be viewed above water through acrylic panels on two sides of the enclosure, as well as from an upper viewing deck that overlooks the exhibit (see Figure 9). Prior to the implementation of Experiment 2, a new haul-out space was added on the right side of the exhibit that allowed another area for the animals to land rest. An acrylic viewing panel allows visitors to easily view the seals when resting on the platform (see Figure 9). Visitors could additionally view the seals underwater through a below-ground viewing acrylic panel that was on a lower level.

#### Data Collection and Procedure

The new haul-out space was completed on 22 February 2022, and Flaherty was moved back into the exhibit from a temporary holding exhibit on 25 February 2022; Experiment 2 started following the completion of the construction at the Northern fur seal exhibit and once Flaherty was back in the main exhibit. Data collection for Experiment 2 ran from 25 February 2022 to 30 April 2022 with a total of 116 observation sessions. Experiment 2 was divided into two conditions: pre-introduction and post-introduction. The pre-introduction condition ran from 25 February to 19 March and consisted of 37 observation sessions. The pre-introduction condition acted as a baseline condition for the existing fur seal (Flaherty) prior to the arrival of the new fur seal. Following quarantine health clearance from the Seattle Aquarium veterinarian and curatorial staff, the two seals were introduced into the primary exhibit together and stayed there permanently together (post-introduction)—unless they had to be moved to another exhibit in order to clean the primary exhibit, in which case researchers did not record data. The post-introduction condition ran from 30 March to 30 April and consisted of 79 behavioral observation sessions. Researchers recorded the behaviors of both seals during this time according to the behavioral ethogram detailed in Section 2.

### 4.2. Results and Discussion

An activity budget for both fur seals displays the percentages that each seal engaged in each of the five classes of behavior following the new animal introduction (see Figure 10). For the northern fur seals, Chiidax’s introduction was associated with a significant increase in Flaherty’s Social behavior (*U*_37,79_ = 1166.00, *p* = 0.006, *d* = 0.33). The introduction was also associated with a significant decrease in Flaherty’s Stereotypic behavior (*U*_37,79_ = 1147.00, *p* = 0.001, *d* = 0.35) (see Figure 11).

Crowd size significantly *increased* following the new animal introduction (*U*_37,79_ = 979.50, *p* = 0.004, *d* = 0.55) (see Figure 12A). For Experiment 2, there was not a significant difference in overall visitor attendance in the post-introduction condition compared to the pre-introduction condition (*t*_53_ = −1.00, *p* = 0.32, *d* = −0.27; see Figure 1B). Visitor length of stay *increased* following the new animal introduction (*U*_37,79_ = 1017.00, *p* = 0.008, *d* = 0.50) (see Figure 12B). Length of stay results approached significance (*p* < 0.05); however, it was not significant under the Bonferroni corrected alpha level for Experiment 2 (α = 0.007; see Statistical Analyses Section 2).

Several impacts were seen in both seal and visitor activity as a result of the introduction. Chiidax’s introduction was associated with a significant increase in Flaherty’s Social behaviors and a significant decrease in Flaherty’s Stereotypic behaviors. Both of these results are suggestive of a positive welfare effect associated with the introduction. There was a significant increase in crowd size at the Northern fur seal exhibit even though there was not a significant increase in overall visitor attendance, which suggests increased visitor interest in the northern fur seals as a result of the new animal introduction. Finally, there was an increase in visitor length of stay that approached significance, which again suggests increased visitor interest as a result of the new northern fur seal introduction.

## 5. General Discussion

### 5.1. Introduction Effects and Experiences

The introduction of Casey into the Harbor seal exhibit had little impacts on the existing harbor seals—Barney and Hogan—or on visitor behaviors—crowd size and length of stay. Neutral introduction effects suggest that the addition of a new seal into the exhibit had little negative welfare impacts on the existing seals [39]. Although there was a decrease in Active behaviors and an increase in Inactive behaviors in Barney, those results are potentially linked to age, which was supported by de Vere et al. [56], who found that older seals in their study had the lowest number of interactions with the other seals in an exhibit. Jenkins [55] also found that there are age preferences among harbor seals when interacting socially with one another; the older interacted less with the younger seals in an exhibit.

For the northern fur seals, the introduction of a new animal was suggestive of several positive welfare outcomes as seen by the increase in Flaherty’s Social behaviors and a decrease in his Stereotypic behaviors. This result was expected due to the fact that Flaherty was previously housed alone, therefore Social behavior should increase when a new fur seal was introduced to the exhibit. Nonetheless, the majority of these interactions were positive, with only 2 out of the 35 Social behaviors observed (5.7%) for Flaherty designated as aggressive. Over 90% of the Social behaviors were affiliative conspecific interactions. The fur seals were observed outside of their standard circannual breeding season, which may have been indicative of the lack of aggression that was observed. If Chiidax’s introduction had occurred during breeding season, we may have seen more aggressive behaviors between the two individuals due to their similar age and their natural behaviors to establish dominance within their hierarchical social structure [47,48,49,51,53,55,56].

Additionally, Chiidax’s introduction was associated with a significant decrease in Flaherty’s Stereotypic behaviors, indicating positive welfare. Although it was not statistically significant, it is important to note that Flaherty’s Active behavior levels increased and his Inactive behavior levels decreased post-introduction. These results—along with the increase in Flaherty’s Social behaviors—are supported by Meyer et al. [57] who found that when housed with members of the same sex, Otariidae species display higher activity levels and increased social interactions throughout the day. Thus, the majority of the changes in Flaherty’s behaviors in the post-introduction condition is suggestive of positive introduction effects and may be linked to positive welfare.

Neutral introduction experiences for Experiment 1 suggest that there was not an increased interest in the harbor seals for the visitors as a result of the introduction. Rather, the increase in crowd size seen in the post-introduction condition was likely due to a significant seasonal increase in overall visitor attendance. This result was plausible considering that there were little differences in the harbor seals’ behaviors, therefore we should observe few differences in visitor attentiveness. The findings from Experiment 1 are supported by Godinez et al. [27] and Margulis et al. [26] who similarly found that visitors spent less time at exhibits where animals were less visible or less active.

For Experiment 2, several positive introduction experiences were seen as a result of the new seal introduction. An increase in crowd size in the post-introduction condition was suggestive of positive introduction experiences for the visitors. This may have been partially due to the increased social activities between the two fur seals, and the fact that a new haul-out space was added to the exhibit that was located right next to the acrylic viewing panels, which allowed for visitors to view the seals up-close (i.e., face-to-face). Additionally, visitor length of stay showed an increase that approached significance during the post-introduction condition. There was no significant difference in overall visitor attendance in the post-introduction condition compared to the pre-introduction condition, which suggests that the increases in crowd size and visitor length of stay were a result of visitor interest in the exhibit.

### 5.2. Implications and Importance of Study

There are many similar comparisons between the present study and previous research studies regarding new animal introductions. Rowden [40] found that when housing Bulwer’s wattled pheasants together, the individuals mimicked natural social and breeding behaviors. In pinniped breeding behavior, northern fur and harbor seals can be social—often in terms of competition over females and resting areas—and tend to form groups on haul-out areas [47,49,50,51,52,53,54,55,63]. Therefore, the social housing structure that was seen in the present study was similar to how harbor seals and northern fur seals behave in the wild, which was indicative of positive animal welfare. Additionally, Schmid et al. [38] found that both the previously existing and incoming female elephants displayed increased social interactions with each other and decreased stereotypic behaviors following the introduction of new elephants into the existing herd. This result was similarly seen in the present study, where Flaherty had increased Social behaviors and decreased Stereotypic behaviors following the new seal introduction, which was indicative of positive welfare and positive introduction effects.

The increase in crowd size and visitor length of stay during the post-introduction condition of Experiment 2 at the Northern fur seal exhibit was suggestive of positive introduction experiences. This was likely due to the positive introduction effects seen for the northern fur seals in terms of increased social interactions, which was in line with previous studies that have suggested that visitors are drawn to exhibits where the animals are active [26,27]. Increased visitor interest in the northern fur seals was supported by Luebke et al. [13] who found that positive perceptions were reported by visitors as a result of observing animals participating in active behaviors, as well as being able to have up-close interactions—such as eye contact—with animals in their study. The Northern fur seal exhibit at the Seattle Aquarium allows visitors to be face-to-face with the seals when resting on dry haul-out spaces, possibly adding to positive visitor experiences. Positive introduction experiences also increase the likelihood that the visitors will continue to support the aquarium in the future [2,4,5,7,14,16,19].

### 5.3. Limitations and Future Directions

The quasi-experimental design of the present study represents a one-condition pretest-posttest design, in that only one actual condition was measured (i.e., seal introductions), and for only two periods; before and after the introduction of a new seal. Greater experimental control is needed to assume any directional effects. To properly control for such an introduction, either within-subjects reversals would be required, such as the removal of the new seal post-introduction (ABA design), or a similar exhibit without new seal introductions would be used for comparison (between-exhibit/subjects comparison). Given the cost-prohibitive nature and negative welfare implications of imposing such controls, this was not realistically possible. Nonetheless, we were cautious in this paper to refer to any observed effects as associated with the introductions, rather than necessarily as a result of such a change. Of the few studies that have examined new animal introductions or changes in social housing arrangements in zoos and aquariums, all have suffered similar limitations. Future new animal introduction research, as well as AVI research in general, should look to incorporate experimental manipulations as much as possible, since much AVI research is currently limited to correlational rather than causal claims.

## 6. Conclusions

The present study fills a gap in the literature of animal behavior by examining how new animal introductions impact the animal-visitor relationship within zoo and aquarium settings. Results suggest a generally positive interaction between introduction effects and introduction experiences. It can be concluded that new animal introductions elicit neutral to positive introduction effects and neutral to positive introduction experiences, therefore suggesting positive AVIs. This study can act as a model for institutions that are looking to examine the effects of new animal introductions on exhibited animals and visitors alike. By understanding the effects of new animal introductions on both animals and visitors, researchers can better understand the welfare of animals through their social housing arrangements, the experiences and education of visitors, and the potential animal-visitor interactions that such introductions promote.

## Figures and Tables

**Figure 1 animals-12-02962-f001:**
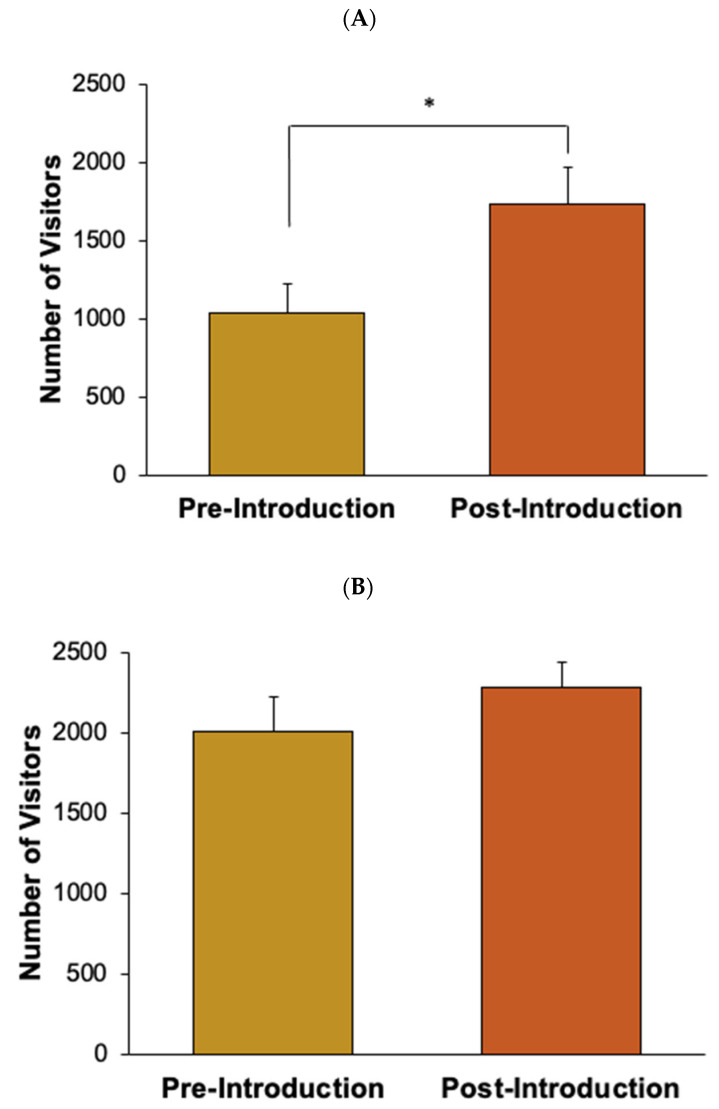
Average visitor attendance (with standard error of the mean bars) to the Seattle Aquarium during the times of observation, comparing pre- and post-introduction. (**A**) shows the mean number of visitors that attended the aquarium during Experiment 1, with standard error of the mean bars. (**B**) shows the mean number of visitors that attended the aquarium during Experiment 2, with standard error of the mean bars. Solid lines and asterisk represent statistically significant results (*p* < 0.05).

**Figure 2 animals-12-02962-f002:**
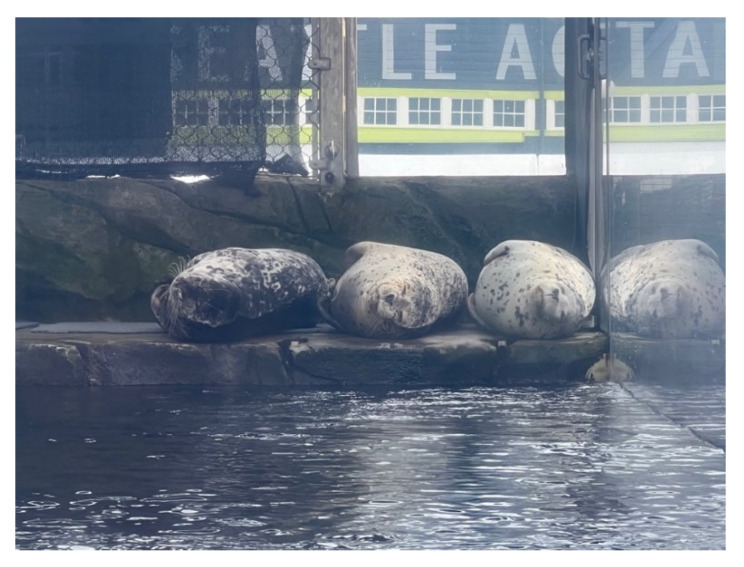
Three harbor seals engaged in Land Resting (from left to right: Hogan, Casey, and Barney).

**Figure 3 animals-12-02962-f003:**
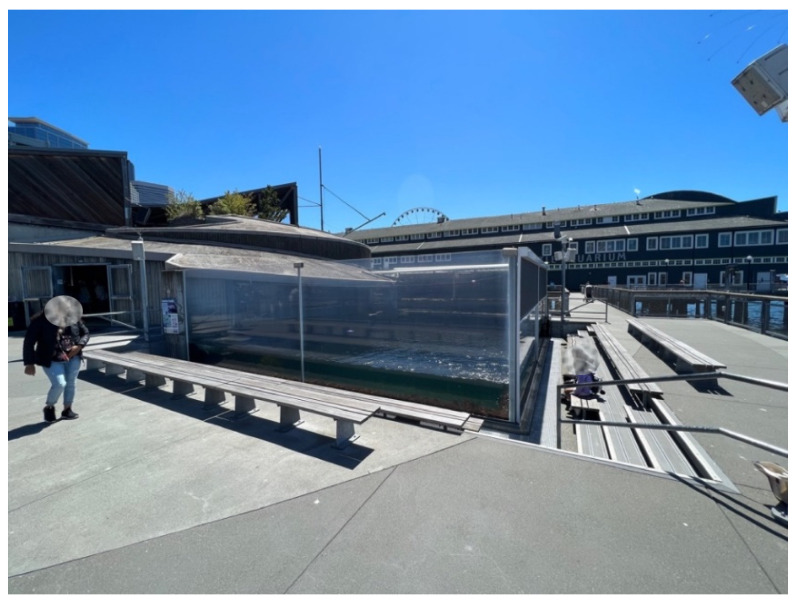
Photo shows the Harbor seal exhibit from the outside viewing area.

**Figure 4 animals-12-02962-f004:**
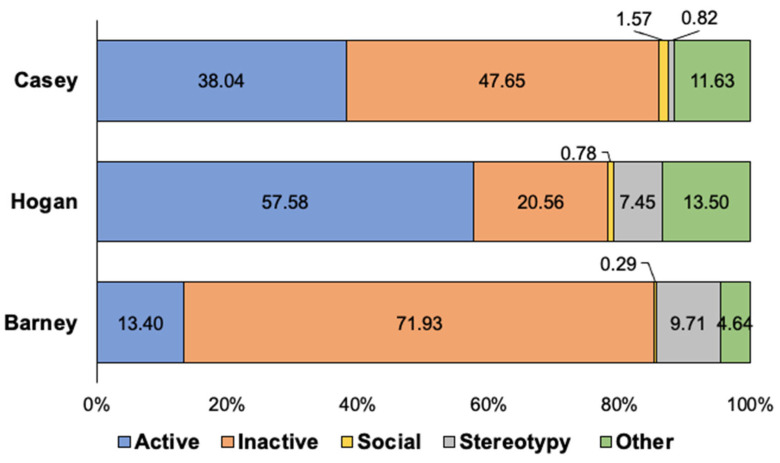
Average percentage of time that each of the three harbor seals spent engaging in each of the five classes of animal behavior (Active, Inactive, Social, Stereotypy, and Other) following the new introduction of Casey (post-introduction). *Note.* As averages, the total count for each seal does not equal 100 percent (~99.71–99.97%).

**Figure 5 animals-12-02962-f005:**
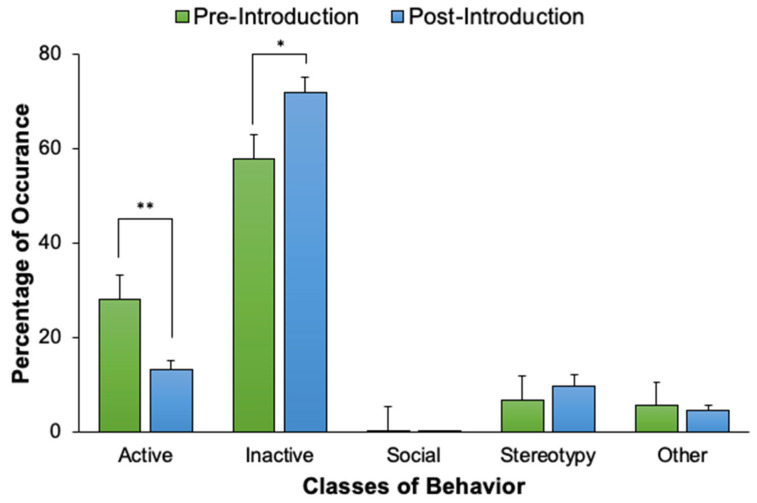
The percentage of occurrence (with standard error of the mean bars) that Barney engaged in each of the five behavior classes, comparing pre-introduction and post-introduction activity levels. Solid lines and asterisks represent statistically significant results in accordance with the Bonferroni correction for Experiment 1 (α = 0.004). * *p* ≤ 0.004. ** *p* < 0.001.

**Figure 6 animals-12-02962-f006:**
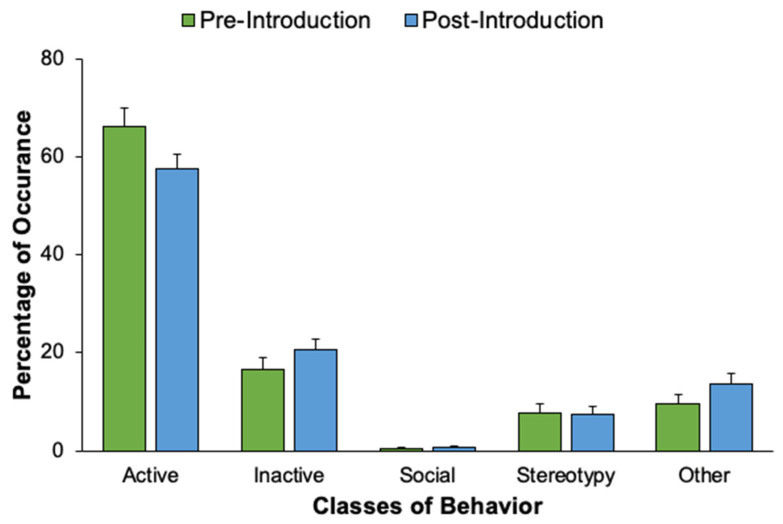
The percentage of occurrence (with standard error of the mean bars) that Hogan engaged in each of the five behavior classes, comparing pre-introduction and post-introduction activity levels.

**Figure 7 animals-12-02962-f007:**
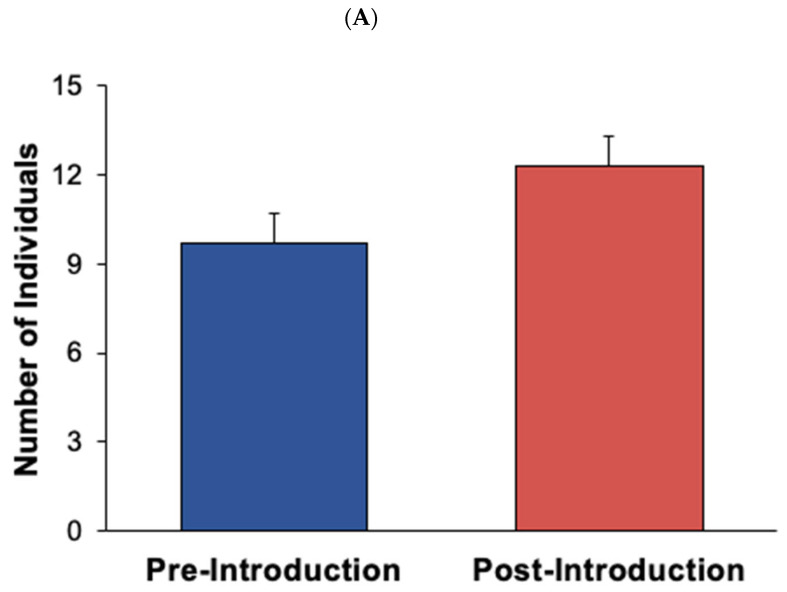
Visitor activity levels for Experiment 1, comparing pre- and post-introduction. (**A**) shows the mean number of individuals (visitor crowd size) in each condition, with standard error of the mean bars. (**B**) shows the average amount of time visitors spent at the exhibit in minutes (visitor length of stay) in each condition, with standard error of the mean bars.

**Figure 8 animals-12-02962-f008:**
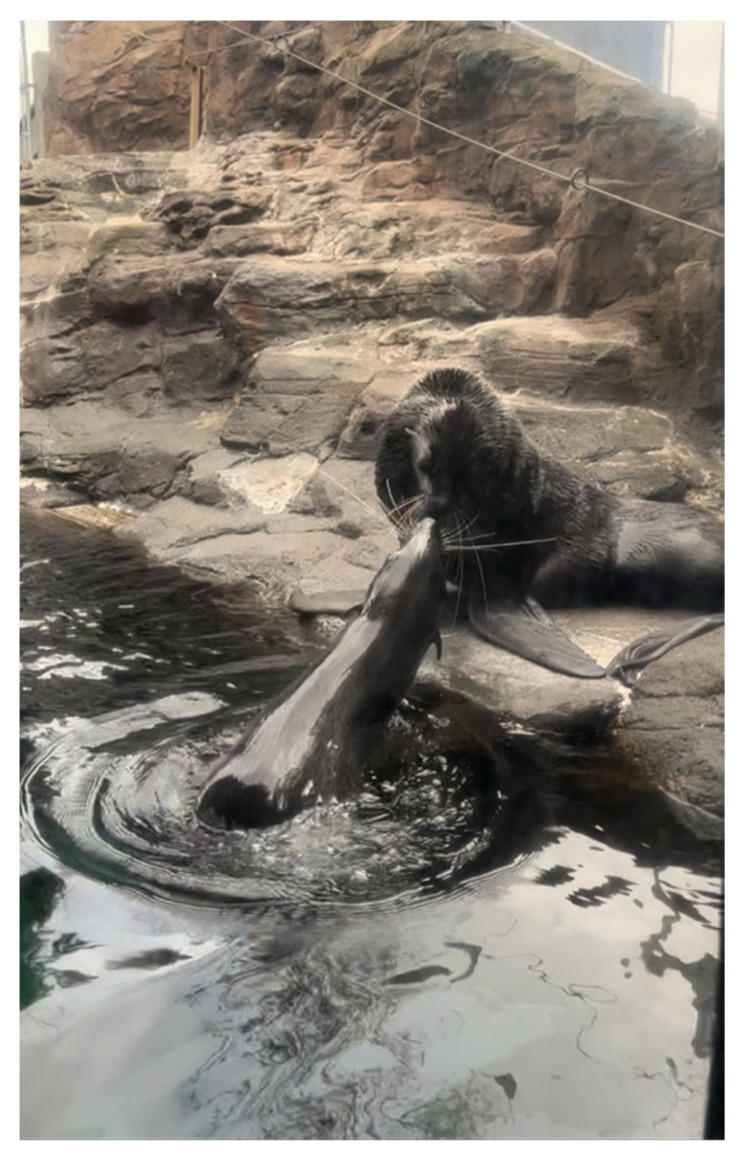
Northern fur seals engaged in Interacting with Another Subject (above: Flaherty, below: Chiidax).

**Figure 9 animals-12-02962-f009:**
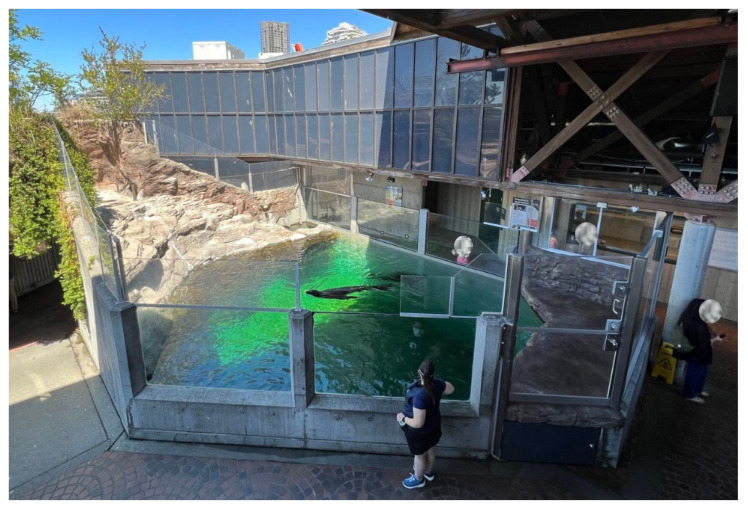
Photo shows an overlook of the Northern fur seal exhibit with one seal engaged in Surface Swimming and one seal engaged in Nonpatterned Swimming.

**Figure 10 animals-12-02962-f010:**
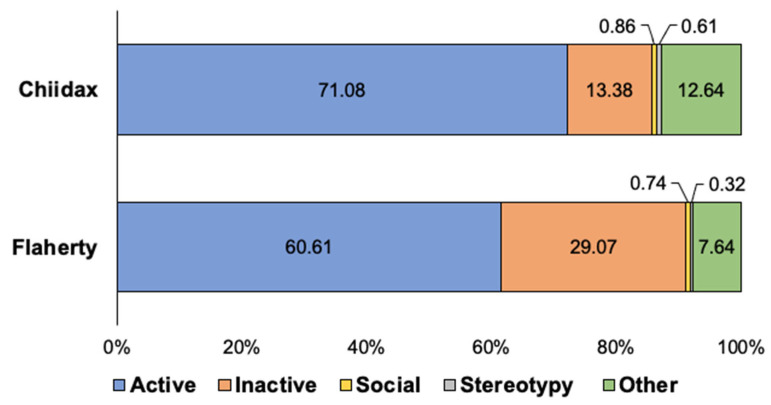
Average percentage of time that both of the northern fur seals spent engaging in each of the five classes of animal behavior (Active, Inactive, Social, Stereotypy, and Other) following the new introduction of Chiidax (post-introduction). *Note.* As averages, the total count for each seal does not equal 100 percent (~98.38–98.57%).

**Figure 11 animals-12-02962-f011:**
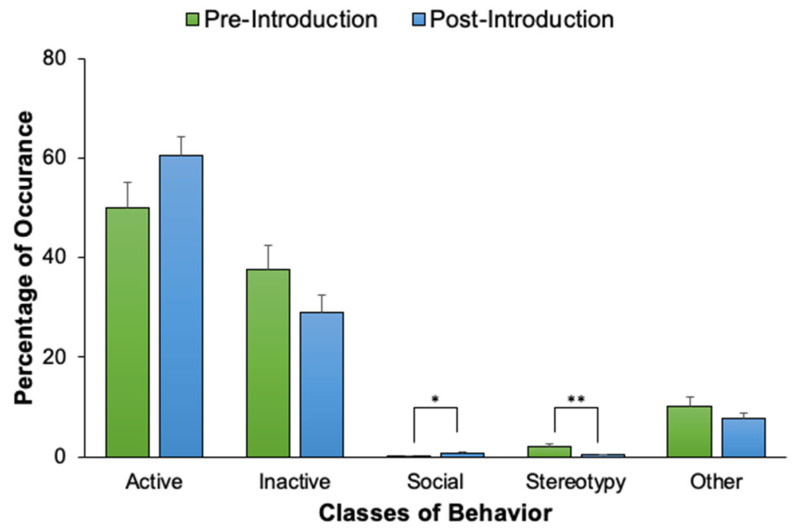
The percentage of occurrence (with standard error of the mean bars) that Flaherty engaged in each of the five behavior classes, comparing pre-introduction and post-introduction activity levels. Solid lines and asterisks represent statistically significant results in accordance with the Bonferroni correction for Experiment 2 (α = 0.007). * *p* ≤ 0.007. ** *p* ≤ 0.001.

**Figure 12 animals-12-02962-f012:**
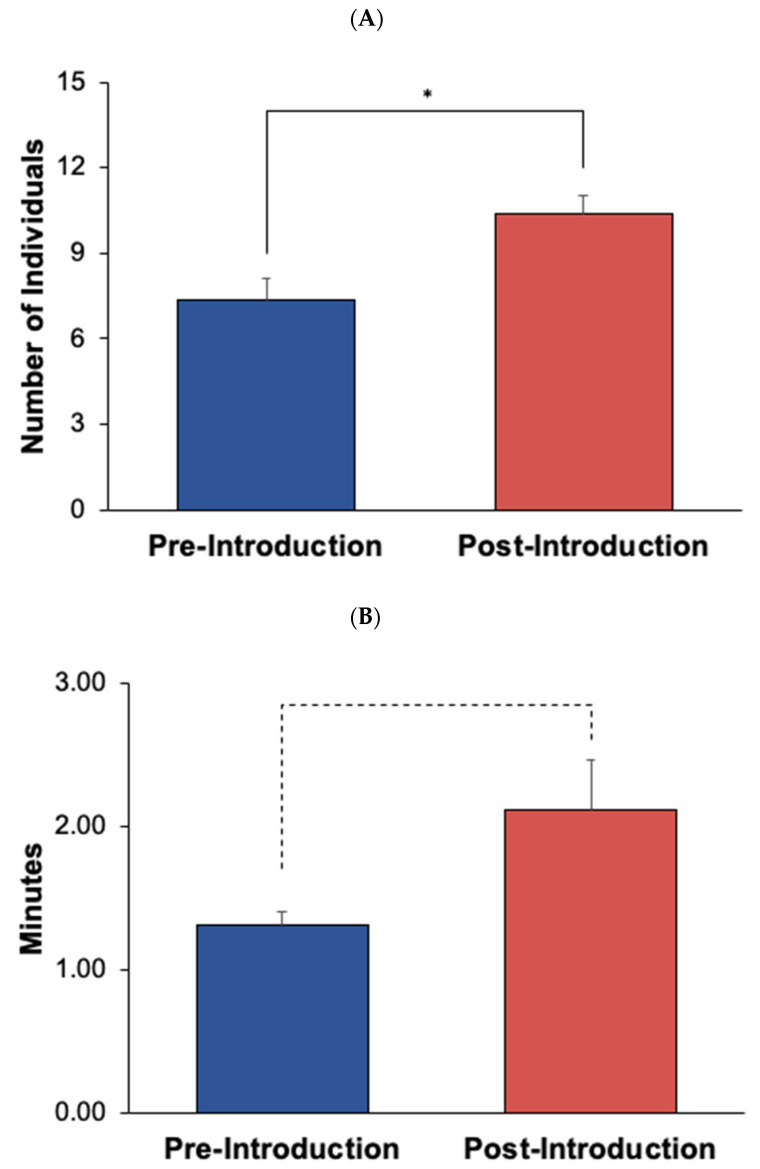
Visitor activity levels for Experiment 2, comparing pre- and post-introduction. (**A**) shows the mean number of individuals (visitor crowd size) in each condition, with standard error of the mean bars. (**B**) shows the average amount of time visitors spent at the exhibit in minutes (visitor length of stay) in each condition, with standard error of the mean bars. Solid lines and asterisk represent statistically significant results in accordance with the Bonferroni correction for Experiment 2 (α = 0.007). * *p* ≤ 0.004. Dotted lines represent near significant results (*p* < 0.05).

**Table 1 animals-12-02962-t001:** Behavioral Ethogram.

Behavioral Class	Terms	Definitions
Active	Nonpatterned Swim (NS)	Swimming underwater without a distinctive pattern or regularity.
	Surface Swim (SS)	Swimming at the surface with some part of body above the water.
	Land Active (LA)	Moving or exploring out of the water using body to travel.
	Foraging (F)	Appetitive, goal-directed behavior, toward food or food-based enrichment, in water or land.
	Other Contact (OC)	Contact with a manipulable object other than a foraging device, such as a non-food enrichment activity.
Inactive	Resting (R)	Subject is not actively moving and appears stationary in water, such as floating or bobbing (can be horizontally or vertically).
	Land Resting (LR)	Same as R, but out of the water.
Social	Vocalization (V)	Using voice to make noises, may be directed at another subject in the enclosure, without aggression.
	Interacting with Another Subject (IS)	Any direct contact with another subject in the enclosure, such as through play without aggression; may include vocalization.
	Aggression (A)	Fighting that includes direct contact (e.g., biting), or no contact (e.g., hissing); may involve IS and V social behaviors.
Stereotypy	Repetitive Swim (RS)	Subject swims in a repetitive pattern for one or more complete rotations.
	Repetitive-Other (RO)	Self-directed repetitive behaviors, including sucking or biting on a flipper.
Other	Grooming (G)	Non-repetitive self-directed behavior that may include rubbing or scratching a body part with another body part or teeth, as well as stretching the neck.
	Out of Sight (OS)	Subject is not visible to the observer; may be off exhibit or hiding.
	Other (O)	Subject engages in a behavior that is not listed above (will describe behavior).

*Note.* Detailed behavioral ethogram of harbor seals (*Phoca vitulina*) and northern fur seals (*Callorhinus ursinus*) was used to categorize the animals’ behaviors.

## Data Availability

Data are available from the corresponding author upon request.

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
