# Peer review of "Effects of New Seal Introductions on Conspecific and Visitor Activity"

_animals, 2022, doi:10.3390/ani12212962_

Round 1

Reviewer 1 Report

I appreciate the laudable attempt by the authors to investigate introductions with an eye to both animal welfare and guest impact. Additionally, I have a personal fondness for the Seattle Aquarium and a great love for Barney. However, I have numerous serious issues with the structure of this paper, the methodology, and the study approach. This paper needs serious overhaul-level revisions to reach the baseline necessary for inclusion in the scientific literature. I have done my best to outline these issues below. Line edits would be too numerous to include here. 

Introduction

1. The entire wording/treatment of AVIs is very confusing to me. AVIs are not inherently positive or negative for animal welfare, yet are treated as such. They are not a methodology, but rather a phenomenon which can be cultivated, adjusted, or informed by research. Also, are you trying to situate this research within AZA-accredited zoos? If so, I think MANY people working at AZA-zoos would strongly disagree with your focus on AVIs as a source of entertainment. 

2. If you're going to discuss AVIs, you HAVE to discuss negative AVIs (of which there are MANY examples). Otherwise, your literature review is biased at best.

3. You have many cases of very generalized statements "close or loud visitors may be detrimental to the welfare of those animals" but almost no specific examples. There is a plethora of literature on this subject, why don't you include it?

4. Co-author Fernandez is definitely an expert on this subject. I personally follow this work quite closely and have a great admiration for it. But there are a number of instances where larger overarching statements are supported by a single Fernandez citation with no other relevant literature mentioned. This seems inappropriate self-citation.

5. Visitor effects are also FREQUENTLY measured through means other than behavioral observations including measures of spatial dispersion, visibility, and hormone (cortisol) fluctuations. These should at LEAST be mentioned.

6. The entire section titled "Animal Introductions and Animal-Visitor Interactions" is seriously problematic. This section asserts multiple times that "only a few studies have attempted to directly examine...animal introductions". This is PATENTLY UNTRUE. There are NUMEROUS studies across multiple taxa. To name just a few Gartland et al., 2018; Abello & Colell, 2009; Burks et al., 2004; Hoff et al., 1996; McCann & Rothman, 1999; Powell, 2010 - the list goes on. This was based on a 5-minute cursory literature search. Social housing (and transitions in social housing) has long-been a significant area of research in zoos and aquariums, but your literature review makes it sound like you're the first one to look at this.

7. Please expand your very short section reviewing pinniped socioecology. There is other literature available to cite. And this gives limited bases of reference for judging the methodology and data presented in your study.

Methods

1. This whole section confuses me. Information is repeated unncessarily and feels scrambled and disorganized. I'll try and list the most major issues.

2. There needs to be more information about the study subjects: sex, age, etc. Names would be helpful too, since they are referenced by name later.

3. What purpose do Figures 1a-1d serve? They feel extraneous.

4. Please cite the "MultiTimer" application properly.

5. Please include specific dates for data collection.

6. What is the data dispersion? How many total hours? How many total observations? What about hours per condition? Hours per animal? How about observations on guests?

7. "conducted 7 days of the week"...what? There are only seven days in the week...

8. Lots of repeated information about methodology

9. The section about how data was collected on people is very confusing. Were there only 6 people sampled per hour? How were adult subjects chosen? There may have been significant bias based on age, whether the subject was in the company of a child/children, etc. This needs to be addressed.

10. Figures are not properly labeled. It is confusing what each figure is even trying to say.

11. Did you apply any sort of randomization? A Bonferonni Correction is not appropriate for these analyses. What about potentially confounding variables like time of day, weather, temperature, etc? Confounding variables NEED to be addressed!

12. Subjects and Setting...is this your results? or is this methods? 

Everything from methods onwards needs to be seriously rewritten and re-organized. Sections for a proper article should be Methods, Results, Discussion. Results and discussion should not be combined. It is difficult to tell what is being referenced where.

Also, why are the conditions referred to as "experiments"? These weren't experiments. An experiment takes place in a controlled environment. You opportunistically studied social transitions, but you didn't control or dictate these transitions.

Overall, this paper needs serious work. It could potentially make a valuable contribution to the literature. However, I highly recommend the authors do more thorough literature reviews, untangle the methodology, and account for necessary confounding variables.

Author Response

Please see the attached file for our cover letter and responses to Reviewer 1's comments.

Reviewer 2 Report

I enjoyed reading this manuscript and only have a few comments that will require minor changes. I have included my comments to the manuscript, in hoping that they will be helpful.

I would however like to warn the authors not to "promote" active interactions between animals and humans (especially when it comes to touching / riding etc.) as there is growing evidence that this negatively impact their welfare. As such, more and more facilities are now moving away from these interactions by promoting passive observations and educational tours. I am referring to sections of the introduction specifically.

Author Response

Please see the attached file for our cover letter and responses to Reviewer 2's comments.

Reviewer 3 Report

I found the topic of this study very interesting. The authors have analyzed the behaviors of the animals before and after the introduction of a new individual in two experiments involving two groups of pinnipeds species. Moreover, they measured the visitor activity, observed as crowd size and visitor length of stay before and after the animal introduction.

I have a few comments, which I have outlined below:

1)      The authors covered much of the recent literature on AVIs. However, if I think about AVI, I think about direct or indirect contact between visitors and animals, in which, in most cases, information about animals, conservation, etc., is given to visitors (e.g., feeding/petting animals; walk-in enclosures; animals shows; etc. As in D’Cruze et al., 2019 definition). In this case, there is an assessment of the visitor presence outside a captive wild animal enclosure. I do not found relevant the introduction on AVI. Instead, I would like to read more about visitor effect and experience, introduction, and pinniped. 

2) My first point may be wrong, however, if more information about enclosure and animal management is provided. In fact, in the Discussion section (lines 492 onwards), the authors stated that visitors can be involved in close interactions with seals. But this information is never given anywhere until discussion (Line 492 and then in line 523!).

3) Moreover, the authors state that this new space was added to the exhibit (Line 491) - but when? before the post-introduction? Or was it also present in the pre-introduction? I think this new enclosure’s feature is important to discuss in detail in relation to the visitor's different behaviors. Correlation or causation?

4) I think the manuscript needs an arrangement in its composition.

- For both experiments, I would move the figures presenting the activity budget (Figure 3 and Figure 7) to the Result section. I find it confusing to have the time budget even before the individuals involved are described.

Instead, start these subsections (experiment 1 line 267 and experiment 2 line 357) with the experiment's aim, the description of the subjects involved, and the "data collection and procedure" section.

Moreover, I suggest moving lines 287 onward, where the harbor seals' enclosure is described, in the "General settings – Subjects and Settings" section (Line 145 onwards), together with the general description and photos of the enclosures. Otherwise, split the "General settings – Subjects and Settings" section (Line 145 onward) in experiments 1 and 2. Same with the description of the northern fur seals’ enclosure (line 382 onward).

I would also like to know the management of Casey and Chiidax before the introduction (where they were before). Were they housed with other individuals? What was the previous composition of the group? Before post-introduction data collection began, did the animals meet for an adaptation period? Why did Chiidax have the opportunity to be introduced into the exhibit without the other individual (line 379) while Casey didn't?

5) I'm not a pinniped expert, but reading the ethogram, I honestly wondered why there are so few social behaviors included. Considering that the study is focused on the introduction of new individuals in pre-existing groups (or in a new enclosure with another individual), I wonder why the social affiliative, aggressive, etc. behaviors have not been analyzed in more detail, as well as the distances between the various individuals (while resting, for example).

I also wonder why authors have not considered performing some enclosure assessment before and after the introduction. In the scientific literature, there are a number of indices that produce a single variability measure of enclosure use, including one, entropy, that is derived from the Shannon index used to assess behavioral diversity.

Finally, 6) Regarding visitors, I have some questions:

- were there information panels around the enclosure where visitors could read information about the animals? If so, what is the average reading time of the panel?

- Was the visitor informed of the new animals in the post-introduction data collection period? If so, how were they informed? At the entrance / near the enclosure?

Author Response

Please see the attached file for our cover letter and responses to Reviewer 3's comments.
